# Relationships between Snowfall Density and Solid Hydrometeors Based on Measured Size and Fall Speed, for Snowpack Modeling Applications

Masaaki Ishizaka[1], Hiroki Motoyoshi[1], Satoru Yamaguchi[1], Sento Nakai[1], Toru Shiina[2], Ken-ichiro Muramoto[3]

[1]Snow and Ice Research Center, National Research Institute for Earth Science and Disaster Resilience, Nagaoka, 940-0821, Japan
[2]Department of Electronics and Computer Engineering, National Institute of Technology, Toyama College, Toyama, 933-0293, Japan
[3]Kanazawa University, Kanazawa, 920-1192, Japan

*Correspondence to*: Satoru Yamaguchi (yamasan@bosai.go.jp)

**Abstract.** The initial density of deposited snow is mainly controlled by snowfall hydrometeors. The relationship between snowfall density and hydrometeors has been qualitatively examined by previous researchers, however, a quantitative relationship has not yet been established due to difficulty in parameterising the hydrometeor characteristics of a snowfall event. Thus, in an earlier study, we developed a new variable, the Centre of Mass Flux distribution (CMF), which we used to describe the main hydrometeors contributing to a snowfall event. The CMF is based on average size and fall speed weighted by the mass flux estimated from all measured hydrometeors in a snowfall event. It provides a quantitative representation of the predominant hydrometeor characteristics of the event. In this study, we examine the relationships between the density of newly fallen snow and predominant snow type as indicated by the CMFs. We measured snowfall density at Nagaoka, Japan where riming and aggregation are predominant, simultaneously observed the size and fall speed of snowfall hydrometeors, and deduced the predominant hydrometeor characteristics of each snowfall event from their CMFs. Snow density measurements were carried out for short periods, 1 or 2 h, during which the densification of the deposited snow was negligible. Also, we grouped snowfall events based on similar hydrometeor characteristics. As a result, we were able to obtain not only the qualitative relationships between the main types of snow and snowfall density as reported by previous researchers, but also quantitative relationships between snowfall density and the CMF-density introduced here. CMF-density is defined as the ratio between mass and volume assuming the diameter of a sphere equal to the CMF size component. This quantitative relationship provides a means for more precise estimation of snowfall density based on snow type (hydrometeor characteristics), by using hydrometeor size and fall speed data to derive initial densities for numerical snowpack models, and the snow-to-liquid ratio for winter weather forecasting. In fact, we found that the method can more accurately estimate snowfall density compared with using meteorological elements, which is the method generally used in current snowpack models, even though some issues remain in parameterisation for practical use. Transferability of the method developed in the

temperate climate zone, where riming and aggregation are predominant, to other snowy areas is also another issue. However, the methodology presented in this study would be useful for other kinds of snow.

## 1 Introduction

The density of newly fallen snow is an important physical property of deposited snow, as it begins to change as soon as the snow reaches the ground. Snowfall density is thought to be determined primarily by hydrometeor types and dimensions in the absence of wind. Some researchers have attempted to study relationships between the density of newly fallen snow and snow crystal characteristics or snow types in falling snow. Power et al. (1964) related snowfall density to snow crystal forms and riming properties within general storm conditions in Canada, and reported relationships between predominate snow crystal type and snowfall density in each snow storm. Snow density values varied, and were distributed over wide ranges, depending on the predominant snow type. Their data showed that the density of snow consisting of dendritic snow crystals was lower than that of plate and column type crystals, and that riming was associated with an increase in density.

 Kajikawa et al. (1989) investigated relationships between snow density and predominant snow crystal types, considering the degree of riming, air temperature, and wind packing. Their results were similar to those of Power et al. (1964) in regard to snow crystal type. Kajikawa et al. (2006) further examined snow crystal types as well as other factors affecting snow density, including the horizontal particle size distribution, contribution rate of graupel, and kinetic energy flux to the snow surface imparted by falling particles. They obtained some important results, including the strong dependence on kinetic energy. However, their observation intervals of about 6 h were so long that the representation of snow type by a single kind of snow crystal must be deemed unsound. Snow types are thought to change during long observation periods, and complex snowpacks may be formed. However, even though Kajikawa et al.'s (2006) results were not consistent, certain aspects of their investigations, for example the contributions of kinetic energy flux and hydrometeor size to density, were very important. To clarify the relationships between snowfall densities and snowfall properties, density measurements for similar types of hydrometeors are required, as well as dimensions.

Recently parameterization of the snowfall density has been a challenging issues associated with the development of numerical prediction models for both snowpack and winter weather forecasting. In the former many models generally used parameterization for the density of newly fallen snow that depend on basic meteorological elements, such as air temperature, surface snow temperature, wind speed, and so on (Lerning et al., 2002: Vionnet et al., 2012: Yamaguchi et al., 2004 ). In the latter, the snowfall density is discussed often relating to snow-to-liquid ratio that is important to accurate prediction of snow height, and is also diagnosed using meteorological data. For accurate estimation of them Milbrandt et al. (2012) introduced bulk density, ratio of  total mass flux to total volume flux of hydrometeors in a snowfall event,  into the bulk microphysics scheme in operational numerical weather prediction. Observations for improving accuracy of estimated bulk density from hydrometeor property and meteorological data also have been carried out (Brandes et al. 2007: Colle et al. 2014). The detailed surface microphysical observation by Colle et al. (2014) also made it clear that crystal habit and riming intensity

were important factors affecting the snowfall density (snow-to-liquid ratio). In any case the snowfall density is discussed throughout relationships with the meteorological data and/or microphysical property of hydrometeors using statistical analysis and/or artificial neural network in some cases (Roebber et al., 2003; Ware et al., 2006). However development of an accurate estimation scheme is currently under way to establish critical relationships between the snowfall density and

meteorological elements including snow properties, because of the diversity of types of solid hydrometeors and complexity of the relation to a snowfall. Quantitative description of a snowfall event is thought to be a key for the snowfall density problem with respect to the diversity of hydrometeors types.

Generally, snowfalls consist of hydrometeors of various sizes, with smaller sizes the most abundant. Ishizaka et al. (2013) presented a new variable to quantitatively describe the main types of snowfall hydrometeors that reflected the contribution of

all hydrometeors to precipitation. In their method, the dominant snow type is represented by a pair of elements, size and fall speed, which are obtained from average size and fall speed, weighted by the mass flux of all measured hydrometeors. This is termed the Centre of Mass Flux (CMF) distribution. Since size-fall speed relationships of hydrometeors are a good representation of particle types, the dominant snow type in a snowfall event may be deduced from the location of the CMF in size-fall speed coordinates. In this work, we characterise the main snow type of snowfalls using the CMF instead of snow

crystal type, and examine quantitative relationships with snowfall density introducing a new variable, CMF-density. We established approximated relationships between CMF-density and snowfall density, which are useful to the accurate estimation of the density of freshly fallen snow in snowpack models, as well as for accurate prediction of snow depth in winter weather forecasting.

In the following section, we describe the methods used to select snowfall events and their classification into categories

representing hydrometeor types, after a brief description of the density observation and measuring system used on falling snow. In Sect. 3, observation results are presented, and the above-mentioned quantitative relationships are introduced together with a discussion of their application to snowpack modeling. A summary is given in Sect. 4, and the appendix describes errors involved in the estimation of snow densification.

## 2 Methods

Observations of snowfall type and density of newly fallen snow were carried in winter seasons between 2013 and 2015 at the Falling Snow Observatory (FSO) of the Snow and Ice Research Center (SIRC) at the National Research Institute for Earth Science and Disaster Resilience (NIED), in Nagaoka, Japan. The FSO is located at 37 °N, 139 °E, in the temperate climate zone, in a coastal area facing the Sea of Japan, where the Siberian monsoon brings heavy snowfalls in winter. A temperature of around 0 °C during many snowfall events generally allows aggregation to be predominant, as well as riming.

## 2.1 Measurement of snowfall density

Density measurements of newly fallen snow were carried out in the FSO low-temperature room, which was kept at about -5 °C. The room has a roof opening (1.2 × 0.6 m) through which snow falls and accumulates on a flat table under windless condition. The density of accumulated snow was obtained by measuring snow depth and weight, using a cylindrical sampler of 10 cm diameter (Fig. 1). Each accumulated snowfall was sampled three or four times by selecting undisturbed areas of snow cover on the table. Snow depth was measured at four points around the sampler for each sample time, using a metal ruler with 1 mm intervals, and the four measurements were then averaged. Snow density, $\rho$ (kg m$^{-3}$), was calculated from the following formula:

$$\rho = \frac{W}{S \cdot h},$$
(1)

where $W$ (kg) is weight of the snow, $S$ (m$^2$) is cross section of the sampler, and $h$ (m) is depth of accumulated snow.

## 2.2 Snow fall observation and determination of predominant hydrometeor type

As the methods used for snowfall observation and determination of predominant hydrometer type were the same as those reported in our previous study on the CMF (Ishizaka et al., 2013), we only briefly describe the procedure here. Snowfall measurements were carried out at the FSO using an automated system, placed in a space enclosed by double-net fences, which detects snow falling under regulated wind conditions with a CCD (Charge-Coupled Device) camera. The size and the fall speed of every detected hydrometeor were extracted from the recorded data using image processing techniques. Particle size is based on the maximum horizontal width of a particle, with a resolution of 0.25 mm. Fall speed was calculated from the vertical displacement of a particle in consecutive frames captured by the CCD camera, at an interval of 1/60 s, giving a resolution of 0.03 m/s. This system is regarded as a disdrometer with a bin size of 0.25 mm and fall speed of 0.03 m s$^{-1}$. Using this system, in a targeted snowfall event we were able to obtain the size distribution and fall speed of hydrometeors corresponding to each bin at 1 min intervals. In this study, we used CMF location to represent the predominant snow type hydrometeor, instead of snow crystal type as used by previous researchers. We have found that the location of the CMF in size-fall speed coordinates provides a good characterisation of predominant snow type. For example, in Fig. 2, the CMF of two snowfall events, A13 and G4, listed in Table 1, are shown along with measured sizes and fall speeds of the falling snow. CMF location varies according to hydrometeor characteristics, i.e. aggregate (A13) and graupel (G4).

## 2.3 Selection of snowfall events

With regard to meteorological conditions, we selected events where air temperature was below 0°C during the entire period of observations in order to avoid cases in which snow melt occurred. As regard wind conditions of observed events, wind speed of graupel type snowfall events were generally greater than that of aggregate ones. The maximum for graupel type events was 8.2 m s$^{-1}$, and wind speed of almost all the graupel events were from about 4 m s$^{-1}$ to 7 m s$^{-1}$. The maximum for aggregate ones was 5.5 m s$^{-1}$, and wind speed of almost all the aggregate events were less than 4 m s$^{-1}$. Wind speed might

somewhat affect aggregation in the air, but it was not so strong in the aggregate cases. Hence wind conditions were not taken into consideration here, because targeted snow for density measurements in snowfall events accumulated under calm air conditions in the cold room.

Even when the observation period was short, 1 or 2 h for almost all events, snow type varied over the event. To examine these variations, CMF was integrated over the whole period, and also over each 1 min interval (1-min CMF). Generally, the 1-min CMF varied with slight changes in either size or fall speed. Therefore, by assessing both the integrated and 1-min CMFs, we were able to select snowfalls with consistent snow, even if changes in size were identified. Cases where different contributing snow types were mixed together in an event, there would be variations in the 1-min CMFs in different areas in size-fall speed coordinates, while the integrated CMF would be located in an intermediate area. There events were excluded from our selection. Figure 3 illustrates CMF distributions for an eliminated and a selected event. In Fig. 3a, the 1-min CMFs are scattered over two areas, corresponding to graupel and aggregate types, and the integrated CMF is located in an intermediate region. In this situation, two different types of snows were falling with almost the same intensity during the event. In Fig. 3b, both the 1 min- and integrated CMFs show the fall of aggregates, with riming below the densely rimed level, of different sizes. This case was selected because the snow type stayed almost the same even though the size of the particles differed. In total, of the 51 events examined, 34 were selected.

### 2.4 Classification of snowfall events

In previous research, identification of predominant snow type was based on snow crystal types in snowfalls. However, our identification is based instead on CMF location in size-fall speed coordinates, which differs with respect to the main hydrometeor types. Hence, we classified snowfall events using four categories – aggregate (A), graupel (G), and two small particle groups (S1, S2) – based on CMF location (Fig. 4). In Fig. 4, the graupel group is separated from the other three by a boundary that represents the size-fall speed relationship for graupel-like snow of lump type as reported by Locatelli and Hobbs (1974). Further, the aggregate and two small particle groups are separated by size. When CMF size was greater than 4 mm, it was classified as aggregate; otherwise it was assigned to one of the small particle groups. The cut-off size of 4 mm was based on the findings of our previous study (Ishizaka et al., 2013), where we found CMFs of aggregate snowfall were larger than 4 or 5 mm in size. Where CMF was smaller than the aggregate group in size and the graupel group in fall speed, we separated small particle groups 1 and 2 based on the boundary that express the size-fall speed relationship for graupel-like snow of hexagonal type. The small particle group S2 (referred to as S2 group), included snowfall events which mainly consisted of graupel-like-snow, between hexagonal and lump types. The small particle group S1 (referred to as S1 group), included snowfall events comprising small particles with lower degree of riming than S2.

**2.5 Estimation of errors in density measurements**

We sampled each event three or four times and averaged the values as mentioned above (Sect. 2.1). Although small error was expected to originate with the scale on the metal ruler used in our observations, the resolution of which was 0.2 mm, larger error might be expected in the readings of accumulated snow depth. From Eq. (1) we estimated the impact of this error on density, $\Delta\rho$ (kg m$^{-3}$), using the following equation,

$$\Delta\rho = -\frac{W}{S \cdot h^2} \cdot \Delta h \,, \tag{2}$$

where $\Delta h$ was set to +/- 1 mm. Eq. (2) indicates that larger errors on density are associated with smaller depths. Therefore, it is preferable to wait until adequate snow depth has accumulated before taking readings. However, the longer the time period the more likely that a variety of snow types will be mixed in one event. Therefore, we restricted the observation period to about 2 h or less, with the exception of two graupel cases (Table 1). Thus in some events, measurements were taken with snow depths of less than 10 mm, with associated calculated density errors of more than 10%. In these cases, error bars were added to graphs to indicate reading errors.

A rather more complicated issue of the accuracy of snow density measurements was expected to be caused by the densification of accumulated snow. However, for a short interval of less than 2 h, the error associated with densification is negligibly small compared with the reading error, as examined in the appendix.

**3 Results and Discussion**

**3.1 Snowfall density: aggregate group and small particle group 1**

Measured and calculated snowfall parameters, including density, for the four different classes of event are listed in Table 1. Figure 5 shows the distribution of snowfall density for each groups, with error bars representing reading errors. In the aggregate and S1 groups, minimum density is 17.9 kg m$^{-3}$ for A3, and maximum is 94.6 kg m$^{-3}$ for A14. Density mainly ranges between approximately 30–70 kg m$^{-3}$, and the maximum is close to 100 kg m$^{-3}$. These values coincide with those reported in Power et al. (1964) and Kajikawa et al. (1989), although the former gives a rather higher density for riming particles. A minimum value of 25 kg m$^{-3}$ is reported in Kajikawa et al. (1989) for an unrimed stellar crystal and 30 kg m$^{-3}$ in Power et al. (1964) for an unrimed dendrite. These minimums are close to our minimum of 17.9 kg m$^{-3}$, which is the smallest of all our observations. Moreover, density values of about 20 kg m$^{-3}$ are also near to that of a single hydrometeor of unrimed or slightly rimed aggregate obtained by 3-D microphotograph analysis, as reported by Ishizaka (1995).

In Figure 6, CMF location (size and fall speed) for the A and S1 groups are plotted with density expressed by grey-scale shading. The figure shows that the event with highest density, A14, also has the highest fall speed, and its CMF is located near the empirical curve for densely rimed aggregate. In contrast, the fall speed of the lowest density event, A3, is below that of moderately rimed aggregate, indicating the low riming of these particles. Event A3 also has the largest CMF size. The CMF for all other events is located at a lower fall speed than that of A14, and at a smaller size than that of A3. Since

aggregate fall speed might be related to riming property, we examined the interrelationship of fall speed, size, and density for the aggregate group, and obtained the following equation:

$$den_{agg} = 82.4v_{CMF} - 6.9d_{CMF} \quad (R^2 = 0.90) \tag{3}$$

where $den_{agg}$ (kg m$^{-3}$) is the density of the aggregate group event, $v_{CMF}$ (m s$^{-1}$) is fall speed, and $d_{CMF}$ (mm) is size. The coefficient of determination is $R^2$, which is defined as the square of the correlation coefficient. Since riming strongly affects particle fall speed, we find from these results that aggregate density depends on both the degree of riming and size. The strong effect of riming, increasing snowfall density, has been emphasised in previous research (Power et al., 1964: Kajikawa et al., 2006: Core et al., 2014). The observation that density decreases as the dimension of an aggregate increases was also reported by Ishizaka (1993) for a single aggregate. When S1 events were added to the regression analysis, a lower correlation resulted; we discuss the S1 cases later (Sect. 3.3.2).

## 3.2 Snow fall density: graupel group and small particle group 2

In our classification based on CMFs, the snowfalls that consisted mainly of graupel were easily selected by the simple criteria previously mentioned (Sect. 2.4). The density of events in the graupel group are shown in Table 1 and Figure 7, as well as in small particle group 2 (S2). Graupel densities range from about 40 to 150 kg m$^{-3}$, and are generally higher than that of the aggregate group. This tendency is understandable because of the strong dependence of density on the riming property as described in the previous section. The highest density in the graupel group was 143 kg m$^{-3}$, which is close to the value of 120 kg m$^{-3}$ reported by Kajikawa et al. (1989).

We also find that the error bars for events in the graupel group are generally larger than for the aggregate group. The reason for this is that snow depth is relatively lower by weight for graupel than for aggregate, since graupel is heavy and a graupel event does not generally last long enough to give a great depth on the ground. As density errors are proportional to weight and inversely proportional to depth, as indicated in Eq. (2), they are generally large in graupel cases.

In Fig. 7, CMF locations (size and fall speed) for G and S2 groups are plotted, with their densities expressed by grey-scale shading. In the figure, the three empirical curves which represent size-fall speed relationships for different types of graupel are illustrated. Lump type graupel has the greatest fall speed, and hexagonal and conical have the lowest and intermediate speeds, respectively. These differences are thought to be due to shape to some extent, but mainly to density differences. A graupel with higher density has a greater fall speed at the same size. Graupel and S2 group densities follow the same trends as the empirical curves instead of rather simple relationship for the aggregate case between fall speed and size expressed with Eq. (3). To show this more clearly, we introduce the distance, *dis*, between the CMFs of the graupel events and the lump graupel curve (solid curve in Fig. 7) as the length of the line drawn perpendicularly to the curve from each CMF. The relationship between the density of the graupel event $den_{grau}$ (kg m$^{-3}$) and the distance *dis* (arbitrary unit) is illustrated in Fig. 8 along with the approximate line expressed as following formula;

$$den_{grau} = 161 - 72.5dis \quad (R^2 = 0.89). \tag{4}$$

Equation (4) shows that density of the graupel group increases linearly as CMF location approaches the curve, indicating higher density for snowfall events consisting of heavier graupels. In determination of the approximated formula (Eq. (4)), four S2 events were excluded. These events showed a similar tendency, but the correlation between density and distance decreased when they were included. S2 case is also discussed later as well as S1 (Sect. 3.3.2).

## 3.3 Snowfall density and CMF-density

In the previous section, we found that the CMF is a quantitative variable which can reasonably explain snowfall density. Equations (3) and (4) present some kind of quantitative relationship between the CMF and density, but they do not cover all cases as we eliminated the small particle groups from the approximation, and the selection of the curve for measuring distance in graupel events was somewhat arbitrary. Therefore, to establish more generally applicable quantitative relationships between snowfall density and CMF-related quantities, we introduce the concept of "CMF-density" in the next section.

### 3.3.1 CMF-density

The CMF is calculated from averaged size, $d_i$ (mm), and fall speed, $v_i$ (m/s), weighted by mass flux, $f_i$, which is defined as a product of mass, $m_i$ (kg), and fall speed, $v_i$ (m/s), as in the following equations,

$$d_{CMF} = \sum_i f_i \cdot d_i / \sum_i f_i \ , \qquad v_{CMF} = \sum_i f_i \cdot v_i / \sum_i f_i \qquad (5)$$

$$f_i = m_i . v_i. \qquad (6)$$

where suffix $i$ represents each hydrometeor in a anowfall event.

Of these, variables $d_i$ and $v_i$ may be derived from observations using our CCD camera or an optical disdrometer, such as Parsivel, and two dimensional disdrometers, but $m_i$ is difficult to acquire. We presented the method for estimating mass flux from empirical relationships in our previous article (Ishizaka et al., 2013). Although the mass flux can be calculated numerically with the method, particle by particle, the mass flux table, which indexes the mass flux calculated in advance for each bin of a given size and fall speed, is used in practice. Fig. 9a is a graphical representation of the table, referred to as a "mass flux chart". From the mass flux table, the mass for each bin can be calculated, and when divided by fall speed, results in a "mass table". Thereby, we are able to obtain the mass for a given CMF size and fall speed. Using the mass, we introduce a new variable "CMF-density".

The CMF-density $den_{CMF}$ (kg m$^{-3}$) is defined as the presumed density calculated by dividing mass $m$ (kg) by the volume of a sphere with diameter $d$ (m), which is equal to a CMF size component for the event, and computed as the following equation:

$$den_{CMF} = \frac{m}{\left(\frac{4}{3}\pi\left(\frac{d}{2}\right)^3\right)} \ . \qquad (7)$$

Using Eq. (7), we can obtain $den_{CMF}$ for each bin, which results in a density table. Fig. 9b shows a graphical representation of the density table, i.e. the "density chart". Using the density table, we can obtain the CMF-density corresponding to the integrated CMF for a snowfall event, and examine its relationship with measured snowfall density. As inferred from the

definition of $den_{CMF}$, the CMF-density can be thought of as the presumed density of the main hydrometeor represented by the CMF assuming a sphere.

### 3.3.2 Relationship between snowfall density and CMF-density

The relationship between measured density (real density) *den* (kg m$^{-3}$) vs. CMF-density $den_{CMF}$ (kg m$^{-3}$) for the aggregate group and graupel groups are demonstrated in Fig. 10. The plots of the two different groups are separated at a CMF-density of about 40 kg m$^{-3}$. For both groups, real density increases with CMF-density. As a first approximation, we fitted a curve for each group using the Levenberg-Marquardt method. A power law function without intercept was adopted, since the curves were thought to start at the origin. Approximate relationships for the aggregate group are expressed as follows:

$$den = 2.5 den_{CMF}^{0.97} \quad (R^2 = 0.71) \tag{8}$$

and for the graupel group,

$$den = 0.34 den_{CMF}^{1.34} \quad (R^2 = 0.92) \tag{9}$$

The fitted curves represent the relationship between real density and CMF-density fairly well in both cases, although there is some scatter. The coefficient of determination $R^2$ and standard error for the aggregate group are 0.71 and 9.7 (kg m$^{-3}$), and those for graupel are 0.92 and 9.2 (kg m$^{-3}$), respectively. For the aggregate group, the coefficient of determination for Eq. (8) is lower than that for Eq. (3), so it might have been better to adopt the latter relationship in the estimation of density. We will examine this issue in a practical process, but for the present we have used Eq. (8) in later calculations.

Moreover, the aggregate and graupel relationships are different from each other. The difference could be related to different packing mechanisms in the accumulation process of the different types of hydrometeors. In the case of aggregates in the air, they have empty spaces that may affect density of accumulated snow, although the spaces might decreases when the aggregate reaches the ground due to deformation or fragmentation of the fragile structure. Since the CMF-density is thought to represent the density in the air in a sense, we could roughly estimate the density for the aggregates on the ground as two or more times as greate as that in the air from the value of the coefficient, 2.5, in Eq. (8) that express as an almost linear relationship between observed and CMF-density. On the other hand, a graupel is thought not to change its density on impact with the ground; the density of a graupel itself in the air directly affects density of the accumulated snow. Moreover, the rather steep dependence of snowfall density on CMF-density expressed in Eq. (9) suggests the effect of particle momentum as Kajikawa et al. (2006) pointed out the effect of kinetic energy. The higher fall speed of high density graupel might more strongly compact the accumulated snow so that the vacant space decreases, resulting in an increase in snowfall density.

In Fig. 11, the densities of S1 and S2 snowfall events are plotted along with the curves fitted for the aggregate and graupel groups. The densities of the two small particle groups plot separately, which indicates that the classification process is reasonable. The S1 group is close to the aggregate curve and the S2 group to the graupel curve. However, the relationship between real densities and CMF-densities is not clear when comparing the aggregate and graupel cases. Observed density for the S1 and S2 groups is lower than that of the aggregate and graupel groups, respectively, at the same CMF-densities. For the

S1 group, the real densities are about 14–50% below the aggregate curve, and for the S2 group they are about from 15–30% lower than that of the graupel curve. Although a number of samples is not enough for a statistical analysis and the correlations are not high, we obtained the following linear relationships between observed density $den$ (kg m$^{-3}$) and CMF-density $den_{CMF}$ (kg m$^{-3}$) for the two groups for later use:

For S1 group,

$$den = 1.6 den_{CMF} \quad (R^2 = 0.77) \, , \tag{10}$$

and for S2 group,

$$den = 1.1 den_{CMF} \quad (R^2 = 0.65) \tag{11}$$

In the small particle groups, snowfall mainly comprises small particles that have a variety of characteristics that cannot be clearly discerned, so that it might be difficult to establish a rigid relationship. Moreover, this uncertainty originates, not only from the variety of snow types, but from uncertainty in the mass flux chart. In the small particles regions, relevant relationships between mass, size, and fall speed have not been established. Thus the mass flux is mainly derived from observations of large particles, which have clear characteristics, and may not be appropriate for small particles. Further research is needed on these targets.

### 3.4 Improving initial density estimation for numerical snowpack models

In current numerical snowpack models, initial snowpack density is generally derived from meteorological parameters such as air temperature, wind speed, etc. The process does not take the type of newly fallen snow into account, even though snowfall density of the main snow types varies widely, as seen in our results. Thus, it is important to introduce a factor that represents snow type into estimation of initial density for snowpack models.

In this study we have presented a method for estimating snowfall density which reflects the predominant hydrometeor in a snowfall event by assessing real time data of both size and fall speed. Application of this method might be expected to improve the accuracy of initial snowpack density for numerical models, as well as the snow-to-liquid ratio for winter weather forecasting. Since snow types change greatly in a short period, it would be better to estimate snowpack density from short interval CMF-densities, for example less than 5 min, as seen in the next section.

### 3.4.1 Comparing estimates of snowfall density derived from CMF and meteorological parameters

In this section we demonstrate the feasibility of using our method to improve density estimation of freshly fallen snow, and compare it with density estimation based on general meteorological parameters. Figure 12 illustrates time series of density estimates derived from both CMF and meteorological elements for an approximately half day period, as well as observed density in 11 snowfall events. Almost all of the events had been eliminated from the list in Table 1 (except A5 and G3) due to complexity in hydrometeor type as mentioned above (Sect. 2.3). Five-minute CMF snow density estimates were obtained using the relationships expressed in Eqs. (8)–(11), after the events had been classified into one of the four categories (section 2.4). The 5 min interval density showed significant variation with hydrometeor type. The density associated with observed

snowfall events, $den_{prd}$ (kg m$^{-3}$), was also calculated for each 5 min interval, using both precipitation amount, $m_j$ (kg m$^{-2}$), and density, $den_j$ (kg m$^{-3}$):

$$den_{prd} = \sum_j m_j / \sum_j \frac{m_j}{den_j} \tag{12}$$

where suffix $j$ represents each interval.

As shown in Fig. 12, calculated density approximately corresponds with observed density. Density estimates from meteorological parameters were derived using the following relationship, statistically derived from measurements in Switzerland as presented by Lehning et al. (2002):

$$\rho_m = 7.0 + 6.5T_a + 7.5T_{ss} + 0.62R_h + 13W_s - 4.5T_a \cdot T_{ss} - 0.65T_a \cdot W_s - 0.17R_h \cdot W_s + 0.06T_a \cdot T_{ss} \cdot R_h \tag{13}$$

where $\rho_m$ is the initial density of new snow (kg m$^{-3}$), and $T_a$, $T_{ss}$, $R_h$, and $W_s$ are air temperature (°C), surface temperature (°C), relative humidity (%), and wind speed (m s$^{-1}$), respectively. In the calculation, we used meteorological data observed at 10 min intervals at our observation site in SIRC. We also recalculated the result excluding wind speed (setting $W_s$ at zero in Eq. (13)). In both estimations, changes in calculated density over time do not mirror the extreme variations in observed density (Fig. 12).

There are some limitations with the comparisons; a precondition of Eq. (13) is that snow exists in the natural environment, but our density observations were carried out in the cold room. However, hydrometeor type does not only depend on the general meteorological elements used in Eq. (13), as identified by rapid, short term changes in snowfall density. Hence, it is difficult to estimate the density of newly fallen snow from these elements only, and it is necessary to introduce information about hydrometer types into the estimation process.

Our method requires the use of additional equipment which can observe hydrometeor size and fall speed in regulated wind speed conditions, for example a disdrometer set in an area enclosed by a net-fence, and a system for calculating CMFs. If these resources are available to use, we can directly calculate from disdrometer data the bulk snow density, which is defined as the ratio between total mass flux and total volume flux for given period, as in Brandes et al. (2007). In fact, we carried out the calculation using 1-min disdrometer data and mass flux, measured with Parsivel and Snow-Rain Intensity Meter (Tamura, 1993), which is a specially designed rain gauge used here,  respectively, for some events, and the results are show in Table 2. In the calculation of volume flux it was assuming each particle was spherical. The ratio between calculated and observed bulk density is given in Table 2. The ration shows remarkable differences between the aggregate and graupel groups, as well as some intra-group variation, indicating that it is related to hydrometeor type. It is also interesting that aggregate ratios, around 0.5, are near to that between estimated density in air and on the ground discussed in section 3.3.2, though we will not consider the issue in more detail here. Throughout, it should be noted that information about hydrometeor type is crucial in the calculation of bulk snow density. Therefore, we consider that our method based on the CMF, which reflects hydrometeor type, is one of the more useful means of estimating snowfall density. In addition, the CMF includes information about particle shape, so that it might be possible to incorporate shape or geometrical factors such as sphericity, dendricity and specific surface area of snow (SSA: Carmagnola et al., 2014) in snowpack models. Moreover, information

about hydrometeor type itself is important for avalanche warning, because some avalanches are induced by particular hydrometeors, such as a large graupel, nonrimed stellar crystal, and very low density snow (McClung and Schaerer, 2006).

The method by which these factors might be parameterised in a numerical model is an important issue, but some difficulties exist. The factors relating to hydrometeor type used here, such as snowfall density, are derived for the initial state of deposited snow, while variables used for estimating density in a general snowpack model, such as air or surface temperature, are not only related to the initial state, but also consecutive states during the discretization interval of the model. On the other hand, hydrometeor type would also certainly affect the process of metamorphism. A snowpack mainly consisting of graupel snow would differently develop from that of aggregate snow. In this way, hydrometeor type also has an indirect influence on the consecutive state of the snowpack, but it is not clear exactly how hydrometeor type affects metamorphism. Further studies are needed on the issues relating to metamorphism.

In this study, the effect of wind on density was not taken into account, though initial density is strongly affected by wind speed throughout the fragmentation of aggregates, changes of kinetic energy and mass flux of hydrometeors, and packing and transportation of snow (Vionnet et al., 2013). Thus, wind has a direct influence on initial density state and is an important factor, but the effect should be considered in the context of hydrometeor type. We have taken a first step by establishing quantitative relationships between density and hydrometeor types without wind. Clarification of wind effect with respect to the hydrometeor type is one of next targets for investigation, along with the effect of melting.

## 4 Summary

In this study, we aimed to establish quantitative relationships between snowfall density and predominant snowfall hydrometeors. We used the integrated CMF (Cntre of Mass Flux distribution), which represented the dominant snow type with a pair of elements, size and fall speed, to characterise the main hydrometeor type in an event, instead of predominant snow crystal as reported in previous studies. In our observations, snowfall events that consisted of almost the same type of snow were selected by assessing both short time interval and integrated CMF. The sampling period for density observation was about 2 h, except for a few events, to avoid densification of the accumulated snow. This short time interval was also favourable for the selection of the similar type snow events. From the observations we developed quantitative relationships between snowfall density and predominant hydrometeors, as follows:

1. Snowfall density ranged from approximately 20 to 100 kg m$^{-3}$ for aggregate snow, and from approximately 40 to 150 kg m$^{-3}$ for graupel snow. These values closely corresponded with those reported by previous researchers.

2. Snowfall density for aggregate snow depended on both the degree of riming and size. Riming was associated with denser snowfall, and higher density of accumulated snow. Density decreased as hydrometeor size increased. Although these trends are similar to previously established results, we were able to quantify the approximate relationship in our study.

3. Snowfall density of graupel snow depended mainly on graupel type, varying between the hexagonal (soft) to lump type (hard).

4. To establish quantitative relationships between observed snowfall densities and snow types (hydrometeors), a CMF-related quantity, CMF-density, was introduced. The CMF-density was defined as the presumed density of the predominant hydrometeors represented by the CMF calculated by dividing its mass by the volume of a sphere, diameter of which is equal to its size.

5. Quantitative relationships between observed density and CMF-density were obtained. The relationships varied with hydrometeor type, which might relate to differences in packing arrangements. Using the quantitative relationships obtained here, it is possible to estimate snowfall density from data on hydrometeor size and fall speed in a snowfall event.

6. Comparison of snowfall density estimates based on CMF (our method) and meteorological elements (a general method using in snowpack modeling) demonstrated that CMF was better at matching changes in observed density, which sometimes fluctuated markedly over short time periods.

These results demonstrate that the CMF method introduced in this study, which combines the two main types of hydrometeors observed in temperate regions, is a reasonable means of establishing quantitative relationships between snowfall density and snowfall characteristics. We also demonstrated the feasibility of using CMF relationships to give an initial density for numerical snowpack models using size and fall speed data from a disdrometer. The method would make it possible to estimate snowfall density for a short term continuously by assessing data derived from an appropriate equipment like our system or an optical disdrometer, for example Parsivel (OTT Hydromet GmbH; Löffler-Mang and Joss, 2000), and two dimensional disdrometer (Kruger and Krajewski, 2002), which automatically measure size and fall speed of hydrometeors, simultaneously measuring their mass by an automated balance, which can precisely measure small amount of snow in a short interval, like Snow-Rain Intensity Meter (Tamura, 1993) used here or Geonor (Bakkehoi and Olien, 1985). It might be also possible to estimate snowfall density from an atmospheric model without these equipment, if the model would be developed to output accurate microphysical properties of hydrometeors that enable to calculate CMF and mass flux.

Moreover, we also highlighted the potential of the CMF method for improving estimation of shape factors, such as sphericity, dendricity, and specific surface area of snow (SSA) for initial snow state in snowpack models, although for practical use further studies are needed to parameterize these factors with integration of other meteorological elements. Among meteorological elements wind effect, which was not taken into account in this study, is one of the most important meteorological elements that affect the initial density of snowpack. The effect of wind should be also considered with respect to hydrometeor types, and clarification of the effect is important issue in the next step. Uncertainty in the relationships for the small particle region is another issue for further investigation, as well as their applicability in other snowy areas. Our observations were carried out at a laboratory located in a relatively warm snowy region, where riming and aggregation are predominant. In colder regions, such as alpine sites, falling snow size is smaller, and aggregation and riming properties may differ. Hence, it might be necessary to establish more accurate relationships for small particles, including a review of the mass-size relationships which affect CMF-density. Although many issues still remain, the methodology presented here offers great potential for estimating the density of freshly fallen snow in both snowpack modeling and winter weather forecasting.

**Acknowledgments**

The authors thank Florent Domine, Co-Editor in chief, for his constructive review and helpful advice. We are also very grateful to two anonymous reviewers for constructive comments and useful suggestion for improving the manuscript. This research is supported by a project of the National Research Institute for Earth Science and Disaster Prevention (NIED)

"Research on advanced snow information and its application to disaster mitigation", and JSPS, KAKENHI Grant Numbers JP26560195 and JP15H01733.

**Appendix**

**Estimation of error in the densification process of accumulated snow**

Kojima (1967) found that a distortion $\varepsilon$ in the thickness of a snow layer, $h$ (m), linearly increases with stress, $\sigma$ (N m$^{-2}$), and

that the relation could also be expressed in terms of density of the snow layer, $\rho$ (kg m$^{-3}$), as follows,

$$\dot{\varepsilon} = -\frac{1}{h}\left(\frac{dh}{dt}\right) = \frac{1}{\rho}\left(\frac{d\rho}{d\rho}\right) = \frac{1}{\eta}\sigma \tag{A1}$$

where $\eta$ is the coefficient of viscosity of the snow.

As the stress, $\sigma$, is pressure induced by accumulating snow on the snow layer, we obtain the following equation;

$$\frac{1}{\rho}\left(\frac{d\rho}{dt}\right) = \frac{1}{\eta}W_{press} \cdot g \ , \tag{A2}$$

where $W$press (kg m$^{-2}$) is the weight of compressing snow in a unit area, and $g$ is the gravity constant. In Endo et al. (1990), for densities of 50–180 kg/m$^3$, $\eta$ is expressed as,

$$\eta = C\rho^4 \ , \tag{A3}$$

where the coefficient, C, varies with snow temperature, crystal type, etc., and has an average value of 0.392. Substituting $\eta$ from Eq. (A3) in Eq. (A2), we obtain the following solution,

$$\rho_t = \left(4\int_{t_0}^{t}\frac{1}{C}W_{press} \cdot g \ dt + \rho_{t_0}^4\right)^{\frac{1}{4}} \tag{A4}$$

Estimating the difference between $\rho_t$ and $\rho_0$ with Eq. (A4), we find that the differences are not large compared with the reading errors discussed in Sect. 2.4 during such short period observations. For example, assuming it snows at a constant rate of 3 mm/h (heavy snowfall) and the resultant density $\rho_t$ reaches 60 kg m$^{-3}$ (a general value), and using Eq. (A4) $\rho_{t0}$ is calculated as 59.4 kg m$^{-3}$ and 58.7 kg m$^{-3}$, over accumulation intervals of 1 h and 2 h, respectively. The differences are a

small percentage of the obtained density, and an order of magnitude smaller than that originated from the reading error mentioned in Sect. 2.4. Therefore, we consider densification errors to be negligible.

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

**Tables**

**Table 1: Characteristics of selected snowfall events, classified into four groups based on snow type, where A is aggregate, G is graupel, S1 is small particle group 1, and S2 is small particle group 2.**

| event | period | density (kgm$^{-3}$) | CMF size (mm) | CMF fall speed (ms$^{-1}$) | sampling snow depth (mm) | sampling weight (g) | CMF-density (kgm$^{-3}$) |
|---|---|---|---|---|---|---|---|
| Aggregate group(A) | | | | | | | |
| A1 | 2013/01/10 10:30–11:30 | 43.8 | 5.5 | 1.0 | 29.5 | 10.2 | 19.9 |
| A2 | 2013/01/10 16:22–17:15 | 34.7 | 6.6 | 1.0 | 37.4 | 10.2 | 15.4 |
| A3 | 2013/01/17 16:18–17:10 | 17.9 | 7.6 | 0.9 | 18.3 | 2.6 | 12.2 |
| A4 | 2013/02/06 08:31–11:05 | 70.6 | 4.2 | 1.1 | 27.5 | 15.5 | 28 |
| A5 | 2013/02/20 13:35–14:10 | 62.9 | 5.8 | 1.1 | 21.1 | 10.4 | 18.7 |
| A6 | 2014/01/09 14:41–15:54 | 40.7 | 4.8 | 1.0 | 9.3 | 3.0 | 22.1 |
| A7 | 2014/01/10 09:45–11:15 | 62.6 | 4.2 | 1.1 | 17.8 | 8.8 | 27.8 |
| A8 | 2014/01/10 16:10–17:10 | 52.9 | 4.5 | 1.1 | 22.8 | 9.3 | 25.9 |
| A9 | 2014/01/17 14:05–15:02 | 42.8 | 6.0 | 1.1 | 19.8 | 6.7 | 17.3 |
| A10 | 2014/03/06 16:20–17:10 | 62.5 | 4.0 | 1.2 | 13.5 | 6.4 | 30.3 |
| A11 | 2014/03/07 09:00–09:55 | 51.8 | 4.1 | 1.2 | 11.0 | 4.5 | 30.3 |
| A12 | 2014/03/07 17:20–18:05 | 55.2 | 4.6 | 1.0 | 20.0 | 8.7 | 23.2 |
| A13 | 2014/03/11 10:07–10:50 | 49.2 | 6.0 | 1.0 | 23.0 | 9.1 | 18.0 |
| A14 | 2014/12/22 11:25–12:05 | 94.6 | 4.4 | 1.4 | 17.0 | 12.1 | 37.0 |
| Graupel group(G) | | | | | | | |
| G1 | 2013/01/18 11:11–14:00 | 77.6 | 4.0 | 1.8 | 3.9 | 2.3 | 51.6 |
| G2 | 2013/01/18 14:03–16:35 | 95.1 | 3.6 | 2.0 | 4.6 | 3.3 | 64.3 |
| G3 | 2013/02/20 13:08–13:28 | 135.1 | 2.9 | 2.0 | 19.1 | 20.2 | 77.3 |
| G4 | 2014/01/10 13:15–14:06 | 143.1 | 3.6 | 2.6 | 25.8 | 28.7 | 89.5 |
| G5 | 2014/01/10 15:13–16:05 | 139.7 | 2.2 | 1.7 | 17.3 | 18.6 | 91.6 |
| G6 | 2014/03/10 12:40–13:40 | 48.7 | 4.1 | 1.5 | 5.5 | 1.8 | 44.2 |
| G7 | 2014/12/17 15:33–16:26 | 100.8 | 2.3 | 1.5 | 19.3 | 14.9 | 70.6 |
| G8 | 2014/12/17 16:31–17:23 | 99.4 | 2.6 | 1.7 | 14.8 | 11.6 | 69.0 |
| G9 | 2015/02/10 09:05–10:05 | 92.6 | 3.9 | 2.2 | 13.3 | 9.6 | 69.4 |
| Small group1 | | | | | | | |
| S1-1 | 2013/02/06 12:32–13:32 | 68.6 | 3.0 | 1.1 | 25.7 | 13.9 | 38.1 |
| S1-2 | 2014/01/17 13:11–14:00 | 73.7 | 3.1 | 1.1 | 24.0 | 14 | 40.0 |
| S1-3 | 2014/02/05 09:22–10:30 | 42.8 | 3.9 | 1.0 | 16.7 | 5.5 | 28.9 |
| S1-4 | 2014/02/05 10:40–11:40 | 33.2 | 3.8 | 1.0 | 9.3 | 2.4 | 28.9 |
| S1-5 | 2014/02/05 11:50–13:15 | 32.9 | 3.8 | 1.0 | 10.3 | 2.6 | 29.2 |
| S1-6 | 2015/02/09 16:33–17:00 | 57.2 | 3.8 | 1.1 | 16.5 | 7.4 | 29.6 |
| S1-7 | 2015/02/09 17:19–17:40 | 55.4 | 3.4 | 1.1 | 19.3 | 8.1 | 35.2 |
| Small group2 | | | | | | | |
| S2-1 | 2014/01/09 15:58–16:55 | 57.8 | 2.9 | 1.3 | 6.3 | 2.4 | 59.0 |
| S2-2 | 2014/02/04 14:12–15:15 | 69.0 | 2.3 | 1.1 | 13.5 | 7.1 | 59.5 |
| S2-3 | 2014/03/08 09:50–11:00 | 74.4 | 2.3 | 1.2 | 5.0 | 2.9 | 72.4 |
| S2-4 | 2015/01/28 08:42–09:40 | 86.5 | 2.4 | 1.3 | 14.3 | 8.9 | 71.2 |

**Table 2. Selected snowfall events and bulk snow density calculated from disdrometer data. Bulk snow density was calculated from total mass flux divided by accumulated volume flux. The ratio between calculated and observed bulk snow density is also shown.**

| event | period (Japan Standard Time) | ① accumurated volume ($m^3\ m^{-2}$) | ② mass flux ($kg\ m^{-2}$) | ③ bulk snow density ②/① ($kg\ m^{-3}$) | ④ observed density ($kg\ m^{-3}$) | ⑤ ratio of bulk to observed density ③/④ |
|---|---|---|---|---|---|---|
| Aggregate group | | | | | | |
| A1 | 10 Jan. 2013 10:30−11:30 | 0.0563 | 1.24 | 22.0 | 43.8 | 0.50 |
| A2 | 10 Jan. 2013 16:22−17:15 | 0.0683 | 1.10 | 16.1 | 34.7 | 0.46 |
| A3 | 17 Jan. 2013 16:18−17:10 | 0.0303 | 0.27 | 8.9 | 17.9 | 0.50 |
| Graupel group | | | | | | |
| G3 | 20 Feb. 2013 13:08−13:28 | 0.0248 | 3.43 | 138.3 | 135.1 | 1.02 |
| G4 | 10 Jan. 2014 13:15−14:06 | 0.0294 | 3.32 | 113.0 | 143.1 | 0.79 |
| G5 | 10 Jan. 2014 15:13−16:05 | 0.0237 | 2.21 | 93.4 | 139.7 | 0.67 |

**Figures**

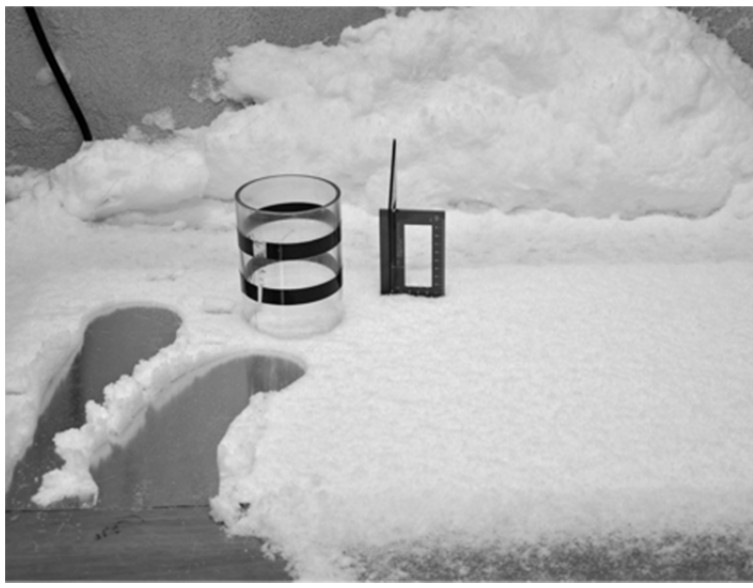

**Figure 1: Photograph of the density observation set up in the low-temperature room, showing accumulated snow on a thin metal plate on the table, and the cylindrical sampler. As an indication of scale, the cylinder is 150 mm in height and 100 mm in diameter.**

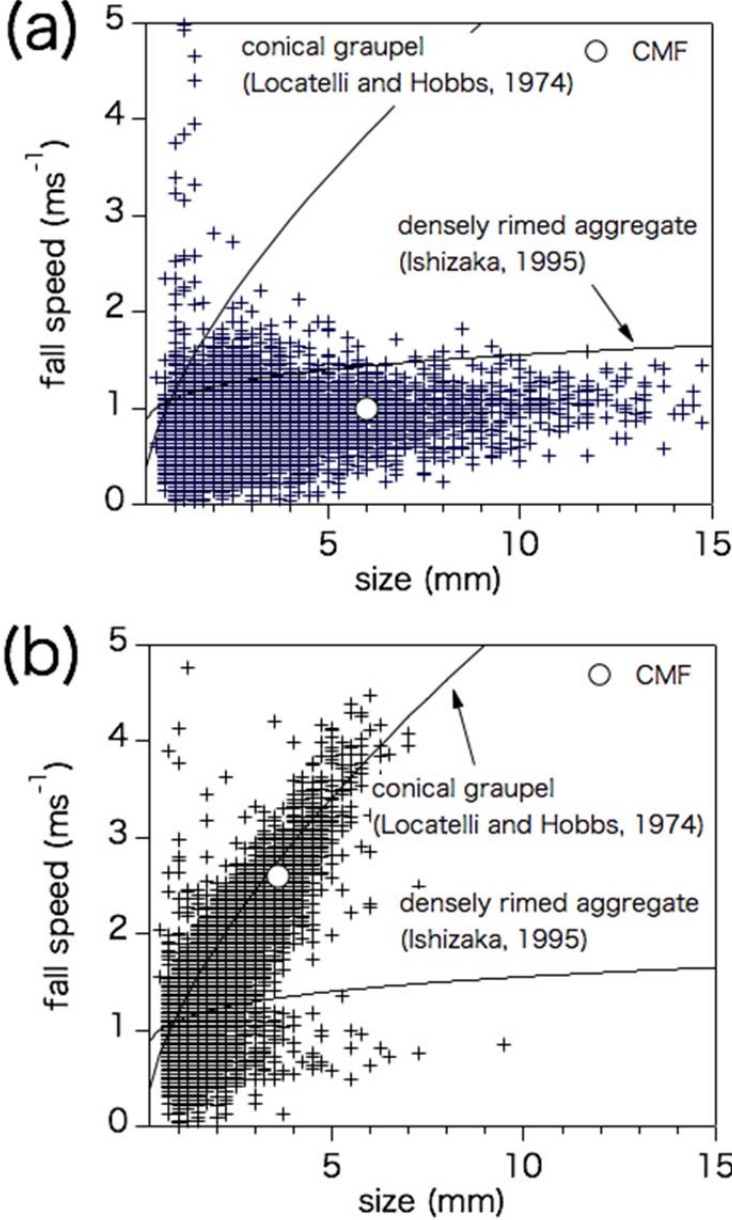

**Figure 2: Distributions of measured size and fall speed (crosses), and integrated CMF (white circle) for different types of snowfall: (a) event A13 (aggregate type); and (b) event G4 (graupel type). Both cases are listed in Table 1. The two curves represent relationships for conical graupel (from Locatelli and Hobbs, 1974), and densely rimed aggregate (from Ishizaka, 1995).**

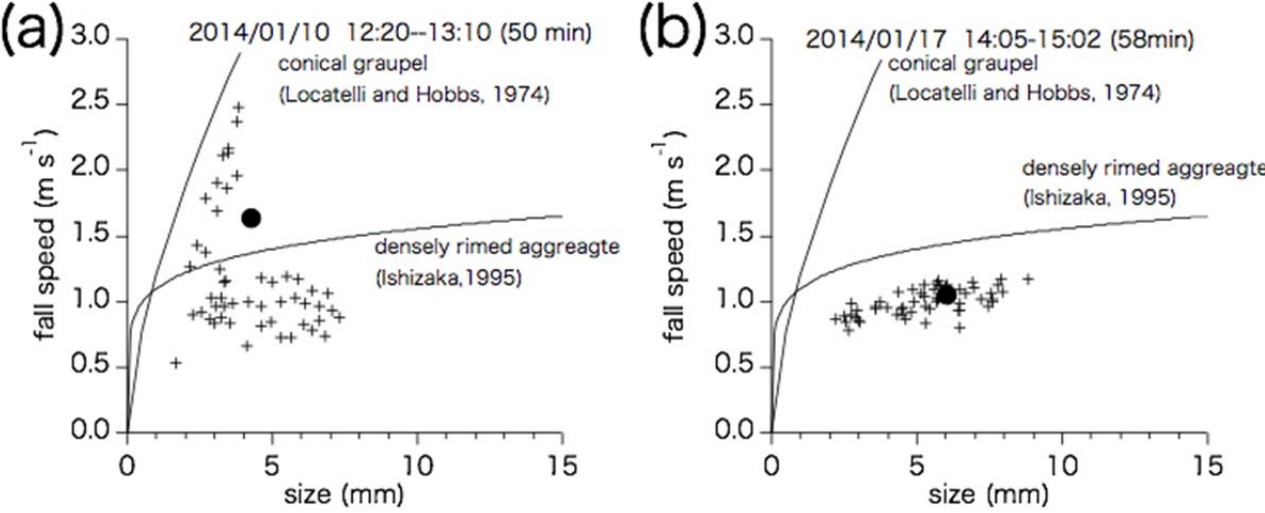

**Figure 3: Distributions of 1-min CMF (crosses) and the integrated CMF (filled circles) for two events: (a) an eliminated event; and (b) a selected event. The two curves represent size-fall speed relationships for conical graupel (from Locatelli and Hobbs, 1974), and densely rimed aggregate (from Ishizaka, 1995).**

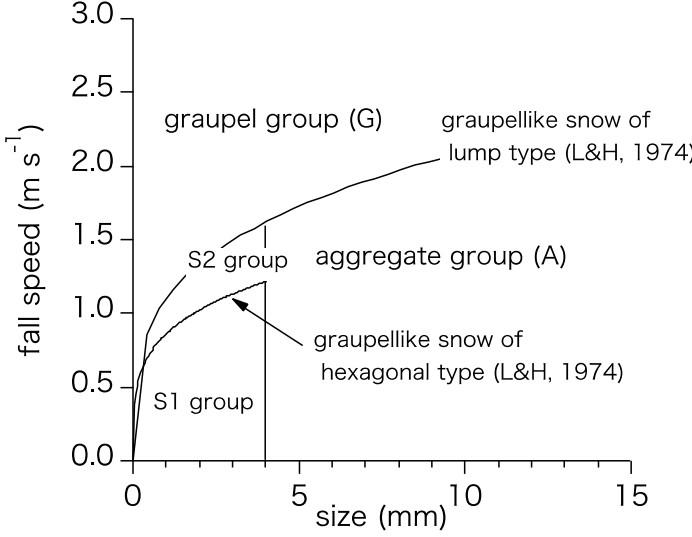

**Figure 4: Categories used in snowfall event classification, showing their location in terms of size-fall speed coordinates. The two curves represent size-fall speed relationships for lump type graupel and hexagonal type graupel (from Locatelli and Hobbs, 1974, denoted as L&H, 1974 on the graph).**

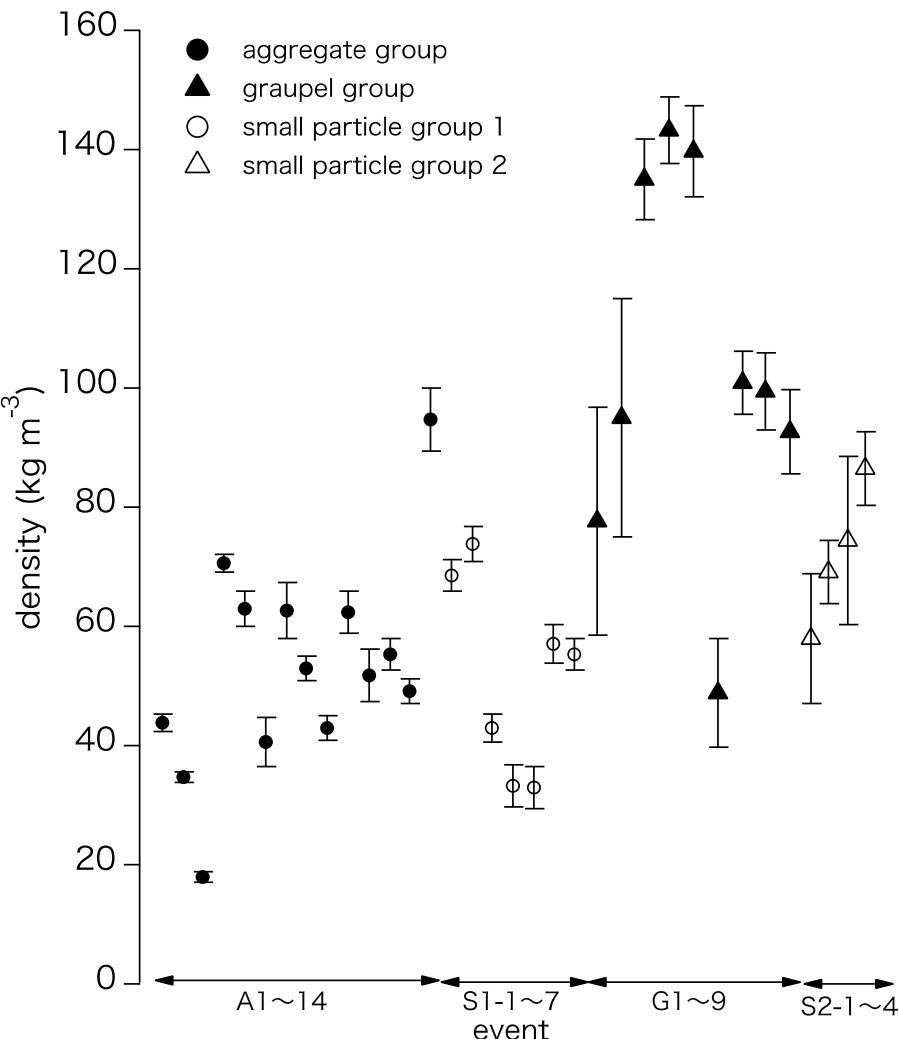

5   **Figure 5: Measured snowfall density of events in each classified group: aggregate (A) group (filled circles); graupel (G) group (filled triangles); small particle group 1 (S1) (empty circle); small particle group 2 (S2) (empty triangle). The error bars indicate reading errors.**

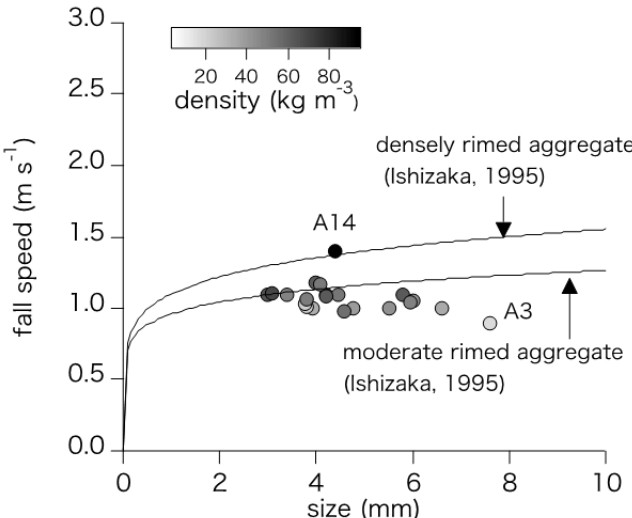

**Figure 6: CMF (grey circles) and measured snowfall density of events in the aggregate group and small group 1. Density is expressed by grey-scale shading in the circles. The two curves represent size-fall speed relationships for densely and moderately rimed aggregates (from Ishizaka, 1995).**

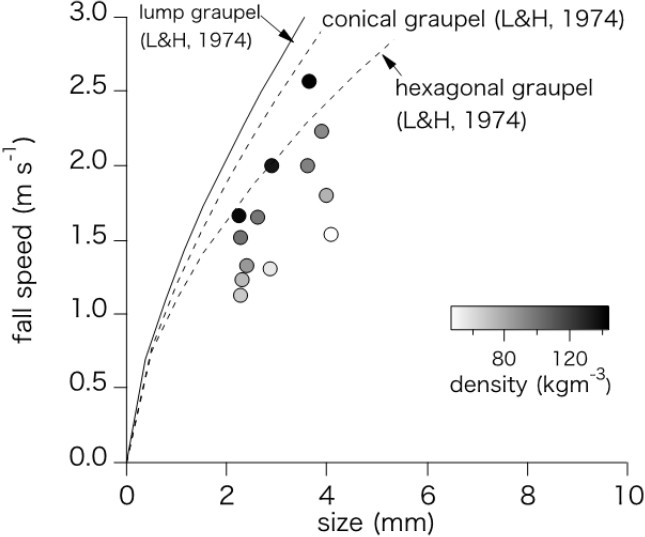

**Figure 7: CMF (grey circles) and measured snowfall density of events in the graupel group and small group 2. Density is expressed by grey-scale shading in the circles. The three curves represent size-fall speed relationships for lump, conical and hexagonal types graupel (from Locatelli and Hobbs, 1974, denoted as L&H, 1974 on the graph).**

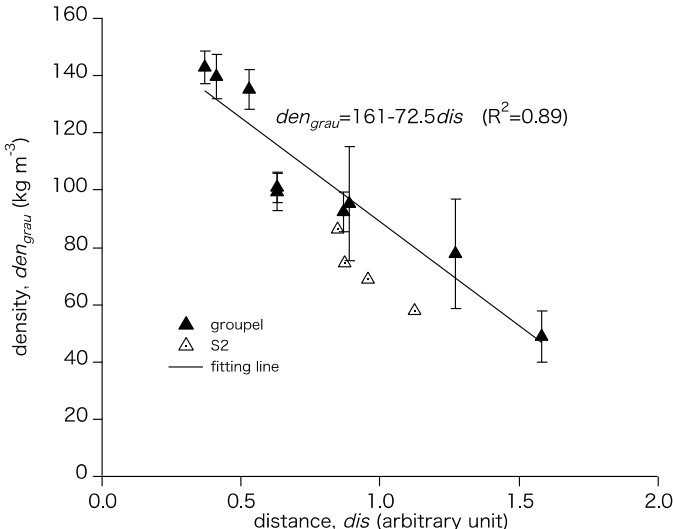

$den_{grau}=161-72.5dis$  ($R^2$=0.89)

**Figure 8: Relationship between snowfall density and distance of graupel group CMF (black triangle) from the fitted curve for lump graupel (illustrated in Fig. 7 by a solid line). The line and regression equation show the approximated relationship. S2 group CMF (empty triangle) is also plotted, but is not included in the relationship.**

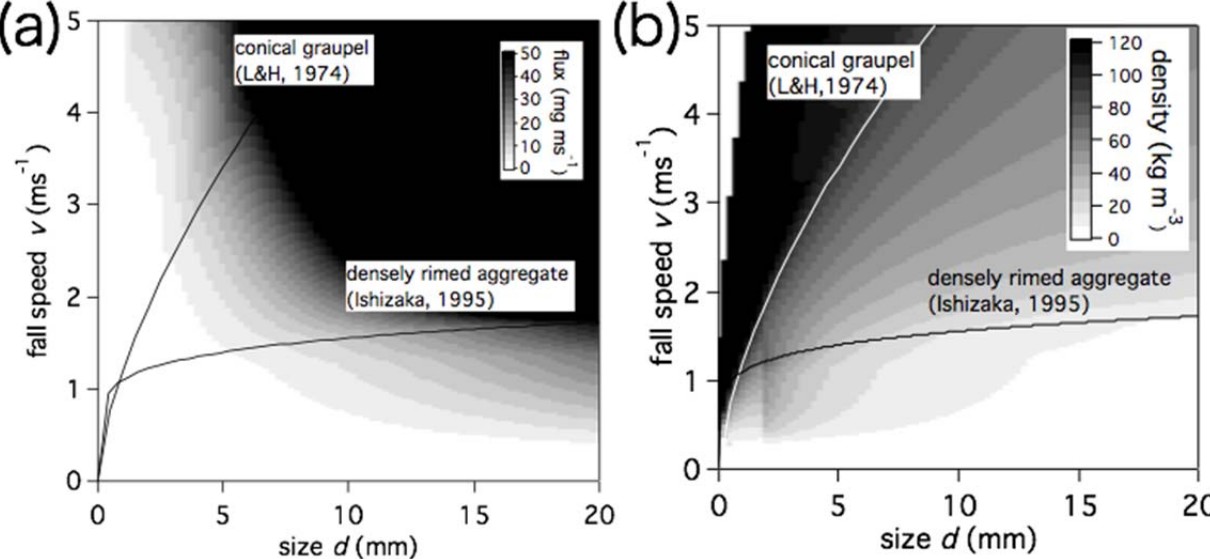

5  **Figure 9: (a) Mass flux chart - a graphical expression of the mass flux table which indexes the mass flux calculated in advance for each bin of a given size and fall speed. (b) Density chart - a graphical expression of the density table that indexes density, calculated by dividing the mass deduced from the mass flux table by the volume of sphere of diameter equal to the bin size. The two curves represent relationships for conical graupel (from Locatelli and Hobbs, 1974, denoted as L&H 1974 on the graph), and densely rimed aggregate (from Ishizaka, 1995).**

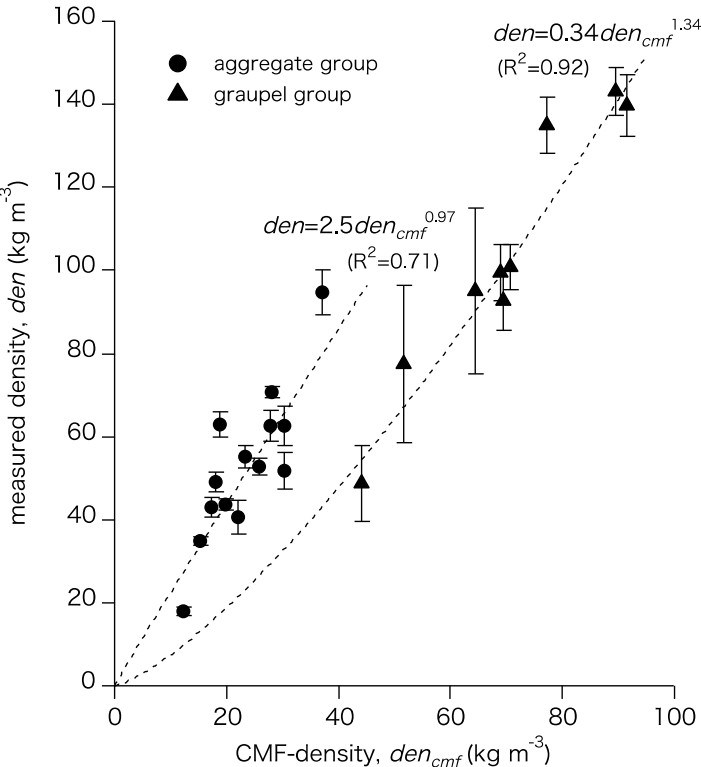

**Figure 10: Relationships between the measured density and CMF-density for the aggregate (filled circle) and the graupel (filled triangle) groups. Dashed lines and equations represent the approximated relationships.**

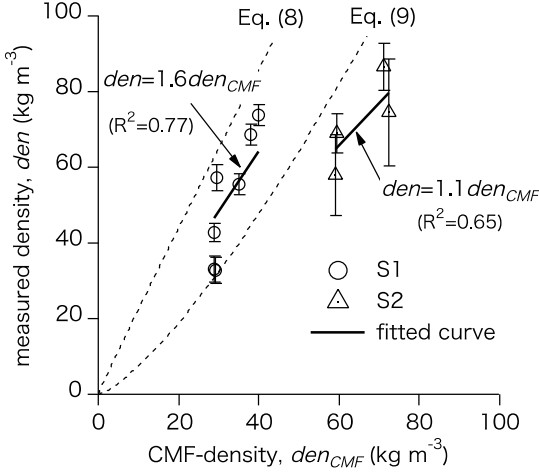

**Figure 11: Relationships between the measured density and CMF density for the S1 (empty circle) and S2 (empty triangle) groups.**
5 **Solid lines and equations represent the approximated relationships. The dotted lines show the same relationships as Fig 10, corresponding to Eqs. (8) and (9).**

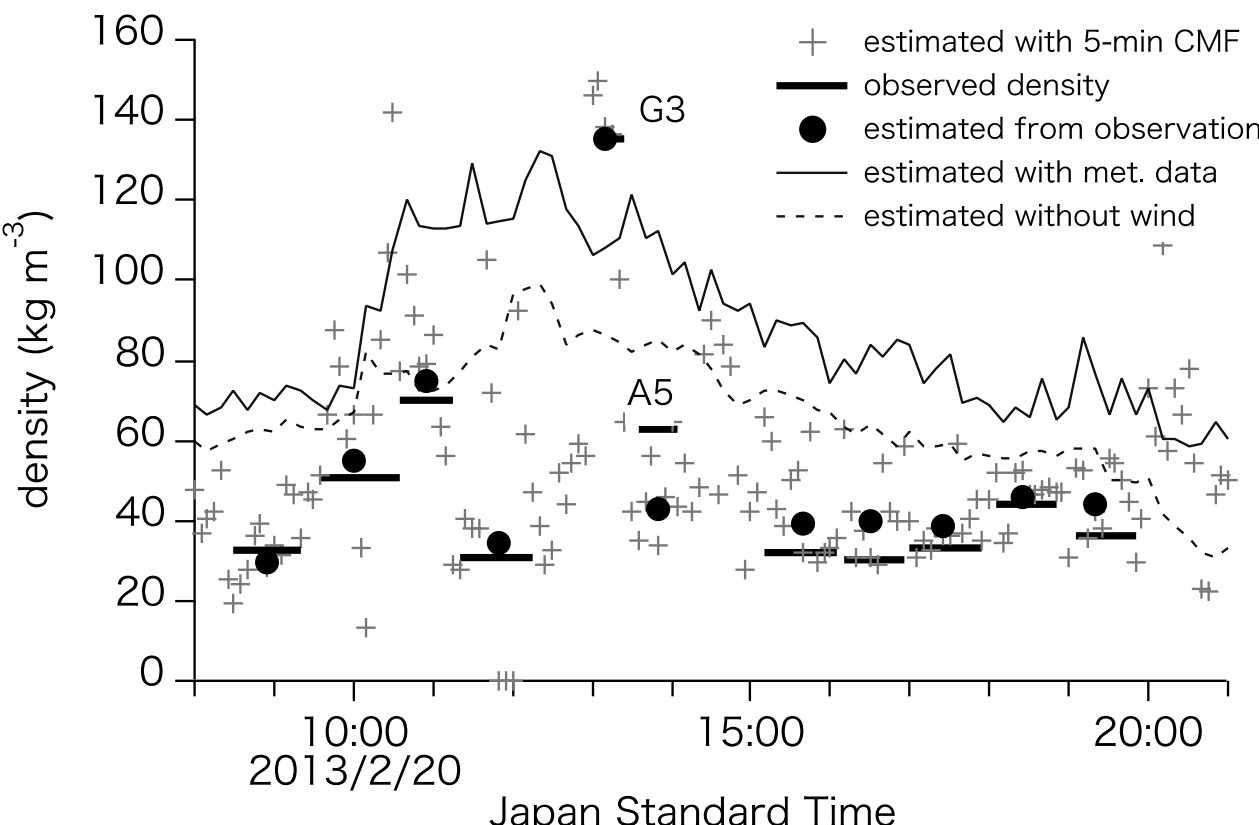

**Figure 12: Time series of estimated and observed density for an approximately half day period. Density observed in 11 snowfall events is indicated with bold horizontal bars, the length of which expresses the time span. Estimated density is indicated as follows: estimated from 5-min CMF (crosses); estimated from 5-min CMFs and mass flux of each 5 min interval for an observation period (filled circle); estimated from meteorological elements using Eq. (13) (solid line); estimated without wind (dotted line). A5 and G3 correspond to events listed in Table 1.**