# Peer review of "Relationships between Snowfall Density and Solid Hydrometeors Based on Measured Size and Fall Speed, for Snowpack Modeling Applications"

_The Cryosphere, 2016_

## Referee Comment (RC1) · Anonymous Referee #1 · 7 Jun 2016

**Review of the paper "Relationships between Snowfall Densities and the Main Types of Solid Hydrometeors Deduced from Measured Size and Fall Speed for snowpack modeling applications" by M. Ishizaka et al. submitted to The Cryosphere.**

This paper studies the links between the density of falling snow and the main type of hydrometeors in a snowfall, which is automatically determined from the measured size and fall speed of every hydrometeor in the snowfall. The authors use a method based on the determination of the Center of Mass Flux (CMF) which has been described in a previous publication (Ishizaka et al., 2013). They restrict their analysis to periods when the snowfall is made of similar hydrometeors (with an air temperature below 0°C). Their results reveal large differences of snowfall density between aggregates and graupel. Then, from the CMF data, the authors compute the "CMF-density" which is related to the density of the main type of hydrometeors. Relationships between snowfall density and "CMF density" are then established as a function of the main type of hydrometeors (graupel or aggregate). At the end of the manuscript, the authors discuss potential impact of their research for snowpack modeling.

Ishizaka et al. addressed an interesting topic for the snowpack modeling community but also for the weather forecasting community since an accurate estimation of snowfall density is necessary to correctly estimate the depth of new snow resulting from a snowfall event. My main comments about this study concern (i) the potential application for snowpack models and (ii) the determination of the "CMF density" and its relationship with the density of accumulated snow on the ground. These questions need to be clarified prior to publication in TC. They are listed below (General comments) followed by more specific and technical comments. Finally, there are too many language and spelling issues, so I strongly suggest an accurate editing by a native speaker.

**General Comments**

1) In the title of the paper, the authors use the expression "for snowpack modeling application" suggesting that this paper could have direct applications for snowpack modeling. However, the authors only include a discussion regarding this point in Section 3.5. Therefore, I strongly recommend extending and improving this discussion to enhance the impact of the manuscript. So far, the expression "for snowpack modeling application" should not be used in the title since it is not accurate. Below I listed several points that could be treated or better explained:

      - as mentioned by the authors, detailed snowpack model used parameterizations for the density of newly fallen snow that depends on air temperature, surface snow temperature, wind speed, … (e.g. Anderson, 1976, Pahaut, 1976, Hedstrom and Pomeroy, 1998, Lehning et al. 2002). If these data are available at their experimental site, the authors could compute the values of density of newly fallen snow given by these parameterizations and compare them with their measurements. It should reveal a large scatter between the parameterized and the measured values that could potentially ne discussed as a function of the main type of hydrometeors (graupel or aggregate)

      - to really illustrate the impact for snowpack modeling, I recommend the authors to derive continuous time series of snowfall density over given time periods. In combination with other atmospheric variables, the authors could use these time series of snowfall density to drive a detailed snowpack model and discuss the impact for snowpack modeling. There is no need to do it over a whole winter but the authors could select a period with successive

snowfall events and present the impact on the simulated snowpack. This would allow the authors to discuss the issues related with changes of snow types during a snowfall event. They suggest (P9 L 16-18) that short time interval (less than 5 min) could be used to derive snowfall densities. Does it work?

     - as mentioned in the abstract the main types of falling snow at Nagoaka consist of rimed particles and aggregates. The author should also discuss in the main text the abilities and limitations of their method for those who want to apply in other regions where different types of crystals in snowfall prevail.

     - finally the authors should mention how their method can be applied practically. Is it only restricted to experimental sites where disdrometers data are available? Can they use their method to develop parameterizations that can be applied in atmospheric models?

2) The "CMF density" (Eq 7, P8) is used by the authors to derive a quantitative value from the CMF data. The authors should clarify the definition of this "CMF density". Is it taken from the density chart (Fig. 8b) at the bin corresponding to $d_{CMF}$ and $V_{CMF}$ (respectively the averaged size and fall velocity of all hydrometeors weighted by their mass flux)?
Did the authors consider other formulations to derive a density computed from disdrometer data? For example, Brandes et al (2007) computed the bulk snow density from 5-min disdrometer data using the ratio between the total precipitation mass and the precipitation volume. The authors could for example compute the ratio between the total mass flux for a given period and the total volume flux for the same period. The volume flux can be obtained assuming a spherical shape for each particle of diameter $d$, as done by the authors to compute the CMF density. Such method has been employed by Milbrandt et al. (2012) to derive the density of falling snow from a cloud microphysical scheme implemented in an atmospheric model.

The "CMF-density" depends on the mass-size relationship chosen for the main types of hydrometeors as described in Ishizaka et al. (2013). Many mass-size relationships can be found in the literature for the different kinds of solid hydrometeors (e.g. Mitchell, 1996; Rasmussen et al. 1999). It would be interesting if the authors could discuss the sensitivity of the "CMF density" to the mass-size relationships used in the CMF method.

The relationship between snowfall density and CMF-density is interesting since it illustrates the influence of accumulation processes on snowfall density. I recommend the authors to extend the discussion regarding accumulation processes since they need to be taken into account to transform a "density" of falling snow in the air derived from a disdrometer into a density of fallen snow on the ground. What are the expected effects for density when aggregates are accumulating on the ground? The authors should also precise the range of usual graupel densities (P8 L23).

**Specific comments**

P2 Introduction: The author should better present in the introduction the need for improvements in the determination of falling snow density: (i) for snowpack modeling and (ii) for wintertime weather forecasting. Concerning this last point, they can for example refer to Roebber et al (2003), Ware et al. (2006) or Milbrandt et al (2012).
The recent study of Colle et al. (2014) presents also a detailed study of the impact of crystal habit and riming intensity on the density of snowfall (related to the snow-to-liquid ratio).

P2-3 L30 L 6: the end of the introduction is not clear and it is hard for the reader to identify the main objective of the study and the structure of the paper. Please consider rewriting the two last paragraphs of the introduction.

P6 L 6: The authors should consider renaming the two categories identified as "small groups 1 and 2". It would help the reader to identify more easily which kind of falling snow particles belong to these groups.

P9 L32: the authors should also precise if there is an expected effect of wind during the snowfall itself; for example the fragmentation of aggregates.

P 10 L 28: as mentioned earlier, with the current version of the manuscript, the authors cannot say that they have shown "the feasibility of using the relationships to an initial density for numerical snowpack model".

**Technical comments**

**Text**

P2 L 15: what the authors mean by "the horizontal size distribution"?

P2 L19-20: which aspects of the study by Kajikawa et al. (2006) are important according to the authors?

P3 L10: please provide the detailed location of FSO in terms of latitude and longitude. A map showing the location of the experimental site would help the reader.

P3 Eq (1) please precise the units of the variables used in this equation. This remark is general and concerns the other equations in the paper.

P5 L 10: please add a reference when mentioning Eq. (3).

P5 Eq (4) and Eq (5): the formulations of Eq (4) and (5) are erroneous. Indeed, $\eta$ is missing in Eq (4) and (5).

P 9 L 8: please refer to Vionnet et al, 2012 instead of Vionnet et al, 2002

P 9 L21: snowpack models use also the snow specific surface area (SSA) (e.g. Carmagnola et al., 2014) to describe snow microstructure.

P9 L23: please add a reference concerning the influence of the crystal type of snowfall on the avalanche danger.

**Figures**

Fig. 6: the location of A3 is hard to identify on this figure since the color of the point is almost white. Maybe the authors could add a black contour around the point so that it can be more easily identified. This remark should be also considered for Fig. 7.

Fig. 9: precise the correlation coefficient for each regression.

Fig 10: the differences between the two accumulation processes are not visually clear.

**Language and spelling**

There are too many language and spelling issues. I listed some of them below but I strongly recommend an accurate editing by a native speaker.

**Text**

Abstract and rest of the text: the authors often use the plural form of nouns ("snows", "snowfalls", "densities", …). I have the feeling that it is not really necessary. Please check the relevance of using it with a native speaker.

Abstract L29 use "snowpack" instead of "snow pack"

Abstract L29 "practical use"

P2 L21-22: the use of paragraph made of a single sentence is sometimes surprising. Maybe the authors can gather this single sentence with the previous or the next paragraph.

P3 L 8 use "snowfalls" instead of "snow falls"

P3 L 16 "through which snow falls and accumulates"

P 3 L27 double use of "falling"

P 5 L 8: the formulation "rather complicated situation" should be rephrased. Snow compaction is not a "situation" that occurs from time to time but a process that occurs as soon as snow accumulates on the ground.

P 6 L 10-11: use "the size component" instead of "a size component"

P7 L 21-22: the sentence "It is found …. for a event has" is complicated and hard to understand. The authors should rephrase it;

P 9 L 6-7: use "snowpack" instead of "snow pack" (same for P9 L 21)

**Figures**

Caption of Fig. 9: add a space between kg and m$^{-3}$

**References (not included in the submitted manuscript)**

Brandes, E. A., Ikeda, K., Zhang, G., Schönhuber, M., & Rasmussen, R. M. (2007). A statistical and physical description of hydrometeor distributions in Colorado snowstorms using a video disdrometer. *Journal of applied meteorology and climatology*, *46*(5), 634-650.

Carmagnola, C. M., Morin, S., Lafaysse, M., Domine, F., Lesaffre, B., Lejeune, Y., ... & Arnaud, L. (2014). Implementation and evaluation of prognostic representations of the optical

diameter of snow in the SURFEX/ISBA-Crocus detailed snowpack model. *The Cryosphere*, *8*(2), 417-437.

Colle, B. A., Stark, D., & Yuter, S. E. (2014). Surface Microphysical Observations within East Coast Winter Storms on Long Island, New York. *Monthly Weather Review*, *142*(9), 3126-3146.

Milbrandt, J. A., Glazer, A., & Jacob, D. (2012). Predicting the snow-to-liquid ratio of surface precipitation using a bulk microphysics scheme. *Monthly Weather Review*, *140*(8), 2461-2476.

Mitchell, D. L. (1996). Use of mass-and area-dimensional power laws for determining precipitation particle terminal velocities. *Journal of the atmospheric sciences*, *53*(12), 1710-1723.

Rasmussen, R. M., Vivekanandan, J., Cole, J., Myers, B., & Masters, C. (1999). The estimation of snowfall rate using visibility. *Journal of Applied Meteorology*, *38*(10), 1542-1563.

Roebber, P. J., Bruening, S. L., Schultz, D. M., & Cortinas Jr, J. V. (2003). Improving snowfall forecasting by diagnosing snow density. *Weather and Forecasting*, *18*(2), 264-287.

Ware, E. C., Schultz, D. M., Brooks, H. E., Roebber, P. J., & Bruening, S. L. (2006). Improving snowfall forecasting by accounting for the climatological variability of snow density. *Weather and forecasting*, *21*(1), 94-103.

---

## Referee Comment (RC2) · Anonymous Referee #2 · 16 Jun 2016

In this study the authors establish a quantitative relationship between the density of freshly fallen snow and the main type of hydrometeors. Hydrometeors are classified trough measurements of their fall speed and size using the Center of Mass Flux (CMF) criteria as defined by Ishizaka et al. (2013). From this relationship the authors introduce a new quantity, the so called "CMF-density", which they relate to the measured density of the freshly fallen snow for aggregate snow and graupel. Finally, the authors discuss potential use of this relationship for snowpack modeling. With this analysis the authors investigate a very interesting approach of estimating snow density of freshly fallen snow, which has the potential to improve current snowpack models. The relationship might further be used to better forecast new snow height of snowfall events, as snow height is strongly dependent on the density of the freshly fallen snow. I have two general comments on the paper, the first concerns the relationship between snow density and "CMF-density" and the second concerns the application of the relationship for snowpack modeling. These general issued should be assessed in advance of publishing in The Cryosphere. Furthermore, I list specific comments and technical corrections, which can help to improve the scientific quality of the paper. Additionally, I strongly recommend to improve figure captions and the English with assistance of a native speaker.

**General comments**

1. The authors establish the relationship of the density of freshly fallen snow to "CMF-density" based on 14 and 9 snowfall events for the aggregate group and the graupel group, respectively. The relationships are thus based on a rather small sample of snowfall events. The authors state "The curves fairly represent the relationship between real density and "CMF-density", although the values scatter around the curves to some extent." (P8 L18-19). However, uncertainty bounds of the curves are lacking and thus, this statement is rather vague. It is important to get a justification of the robustness of the curves. A possibility to establish the uncertainty of the relationships is to perform a "leave one out cross validation" on the data. Another possibility would be to validate the results by a different sample of measurements for aggregate snow and graupel, respectively. However, the second might be difficult to achieve as only a certain amount of measurements may be taken during one season.

2. The title of the paper has the reader waiting for an application of the established relationships for snowpack modeling. The aspect of the applicability of the rela-

tionships for snowpack modeling is however only discussed in a perspective way for possible applications in the future. The authors mention that the relationships between "CMF-density", which may be determined from fall speed and size measurements of hydrometeors, and the density of freshly fallen snow may provide an improvement on the estimation of snow density in current models, where the initial density of snow is modeled based on meteorological conditions, while the type of hydrometeors is not considered. As the paper title states "..., for snowpack applications" I would very much recommend that the authors show a comparison of the performance and improvements of the new method of snow density estimation compared to currently employed methods in numeric snowpack models.

**Specific comments**

- It is informative how you derive the uncertainty of the error in your density estimation based on the reading error. However, you do not mention the uncertainty of your scale, which might have a similar impact as the reading error depending on the accuracy/uncertainty of the scale. If this uncertainty is negligible, please state.

- In section 2.4 "Estimation of errors in density measurements" you further give the derivation how you estimate densification of the snowpack, even though you finally state that for your application densification is negligible. Your paper would benefit by moving the whole derivation (P5 L10 -24) to the supporting material. Furthermore, equations (4) and (5) have to be revised.

- The first section of the results 3.1 "Classification of snowfall events" states how you classify your groups. In my opinion this should go to the methods section, as this section gives your methodology how you separate the groups but no results.

- The analysis is restricted to snowfall events lasting about 1-2 hours (allowing for densification of the snowpack to be neglected) and to two snow types: aggregate snow and graupel. A discussion about the applicability of the method for different snow types and longer lasting events would be interesting.

- Figure Captions: all of your figures are sparsely described. Please, give more specific figure captions! Furthermore, I recommend to that figure captions are autonomous, i.e. define abbreviations. Example: "Figure 2: The distributions of measured sizes and fall speeds (crosses), and the integrated CMFs (white circle) of different types of snowfalls. a) corresponds to event A13 (aggregate type) and b) to event G4 (graupel type). Both cases are listed in Table 1. The two lines show the relationships of size and fall speed for conical graupel as described by Locatelli and Hobbs (1974) and densely rimed aggregate as described by Ishizaka (1995)."

- When you refer to previous sections specification of section is helpful for the reader.
  - e.g. P6 L19: "... originating from the reading errors (Section 2.4)."
  - e.g. P7 L8: "... criteria previously mentioned." -> "... criteria mentioned in Section X.X.", X.X = number of section.

**Technical corrections**

- P1 L31: plactical -> practical

- P2 L21-22: Try to eliminate one-sentence paragraphs.

- P3 L12: "The winter temperature, around 0°C..." Is this the mean winter temperature?

- P3 L28: "CCD" Please define this abbreviation.

- P8 L25: "SI" -> S1

- P10 Summary: The summary should be autonomous and abbreviations should be defined.

- I recommend to rephrase the following sentences to make them more precise:

  - P4 L19: "If different snow types, ..."
  - P7 L21: "It is found that the densities..."

- Missing/spare spaces:

  - P1 L24: "...hydrometeors.As a result..." -> "...hydrometeors. As a result..."
  - P4 L10: "... aggregate type (A13)and graupel..." -> "... aggregate type (A13) and graupel..."
  - P7 L21: "...density is , the larger..." -> "...density is, the larger..."

Language

There are numerous lingual issues. I will mention some which will likely need to be changed. I strongly recommend to improve the English by assistance of a native speaker. Abstract (and later on in the paper): snows -> snow, hydrometeors types -> hydrometeor types, snowfalls -> snowfall events

---

## Author Comment (AC1) · 24 Aug 2016

Thanks for reviews of our manuscript.

We believe that most of reviewers' comments were fair and proper, and we found them very helpful in revising our paper.

We have addressed all the comments and substantially revised the documents and trust that all revisions will be satisfactory.

For our detailed response, we updated a zip file (Ishizaka_revision.zip) including a

response letter and a revised paper to the supplement.

The file name of the responses to each reviewer's comments is "Response letter.pdf". The file name of the revised manuscript is " manuscript_Ishizaka_snowfall_density_R2.pdf".

I look forward to hearing further comments or suggestions.

Please also note the supplement to this comment:
http://www.the-cryosphere-discuss.net/tc-2016-68/tc-2016-68-AC1-supplement.zip

---

## Author Response (AR1)

**Authors' responses to the Reviewer #1**

We would like to express our appreciation to the Reviewer #1 for your constructive reviews and helpful advice.

We have revised our manuscript and corrected mistakes according to your general and specific comments.

The revised manuscript file is "manuscript_Ishizaka_snowfall_density_R2.pdf".

Fist of all, the title has been slightly changed by following the editing process of a native speaker.

The new title is "Relationships between Snowfall Density and Solid Hydrometeors Based on Measured Size and Fall Speed, for Snowpack Modeling Applications"

Our responses to the comments are described as follows:

**Responses to the general comments 1)**
Comparison of estimation of snowfall density by our method with by a method used in current snowpack model

On the reviewer's instruction, we added the new section 3.4.1 to discuss applicability of our method. In the section, we presented comparison of estimated snowfall density using our method with that used in the snowpack model developed by Lehning et al. (2002) and showed time series of observed and estimated densities.

As mentioned in the section, it was hard to satisfy preconditions for the comparison, but the results suggested that information about hydrometeors should be necessary for accurate estimation of snowfall density.

Practical use

In the added section 3.4.1, we also described how practically obtained snowfall density of a event from a short time CMF (5-min CMF) during the event, which can be obtained from the measurement of size and fall speed of hydrometeors. (P10 L25-28; in the revised manuscript: same in below)

We also described additional equipment, which are necessary for practical operation of our method in the summary. (P13 L8-12)

Applicability to other snowy region

We mentioned our idea in the summary. (P13 L18-22)

Applicability in an atmospheric model

We believe it will be capable to apply our method to the atmospheric model if the model could output accurate microphysical properties that enable to calculate CMF. We briefly comment this idea in the summary. (P13 L12-14)

**Responses to the general comments 2)**

Definition of CMF-density

We presented the detailed definition of the CMF-density in the section 3.3.1 using Eq. (5) and Eq. (6).

Bulk density

The bulk density can be deduced from the disdrometer data. We calculated bulk density using the disdrometer data and compared them with the observed density in section 3.4.1. The ratios between bulk and observed density strongly depend on the main type of a snowfall event (P11 L12-24). These results indicate the importance of type of hydrometeors in the snowfall density problem.

Dependence of CMF-density on the mass-size relationship

As the reviewer points out, CMF-density and CMF itself depend on the choice of the mass-size or size-fall speed relationships and so on. We have already discussed these situations in the previous study (Ishizaka et al., 2013). We think it is difficult to determine the best choice of them and it should be practically resolved in case by case, namely, considering the predominant types of hydrometeors in the targeted snowy region where they are used. The problem also relates to application possibilities of the method to other regions, especially to the region where small particles are predominant. Thus we mentioned this problem in the last part of the section 3.3.2 (P10 L4-7), and added the following sentence in the last part of summary.; "It might be necessary to establish…...including a review of the mass-size relationships which affect CMF-density(P13 L21−22)" to describes the application possibilities of our method in colder snow regions, where snowfall may be consisted with small particles and no appropriate mass-size and size-fall speed relationships have not been established,

We think further discussion about this problem is outside of the scope of this manuscript.

Relationship between snowfall density and CMF-density

We think that CMF-density should reflect density of snow in the air because it was derived from data of hydrometeors in the air while snowfall density is measured on the ground. Thus relationship between CMF-density and snowfall density should indicate the relationship between density of snow in the air and that on the ground. We added more detailed discussion on this problem in the section 3.3.2 using the quantitative relationships expressed by Eq. (8) and Eq. (9). (p9 11-21) Moreover the ratio of bulk density for aggregate type snowfall is considered to relate this problem, which we briefly mentioned in the section 3.4.1. (p11 20-21)

Figure 10 in the previous manuscript was eliminated because it could not accurately express the argument given in the revised manuscript.

**Responses to specific comments**

Improvement of introduction

On the reviewer's instruction, we added description on the need for improvements in the determination of falling snow density for both snowpack modeling and winter weather forecasting. We are very grateful to the reviewer for suggesting useful references.

The last two paragraphs of the introduction in the previous manuscript

We deleted them and revised the final part of the introduction.

Categories "small groups 1 and 2"

As the reviewer's suggestion, we also think it is better to rename these categories, but we could not conceive more appropriate abbreviation for them. We only renamed them with a minor change, namely, "small group" to "small particle group", with more detailed description for them in the section 2.4. (P5 L21-23)

Wind effects to snowfall itself

We added the description about wind effects relating hydrometeors to the text. (P12 L4-6)

Feasibility of using our relationships to an initial density for numerical snowpack model

As mentioned in the response to general comment 1), we added more concrete discussion for the applicability of our method to a numerical snowpack model in the section 3.4.1.

**Responses to technical comments**

**Text**

"the horizontal size distribution"

→ "the horizontal particle size distribution" (P2 L15)

which aspects of the study by Kajikawa et al. (2006) are important・・・
The aspects were added as follows:
"for example the contributions of kinetic energy flux and hydrometeor size to density" (P2 L20)

Location of FSO
The longitude of FSO was added.

Units of variables in Equations
We gave units to all variables in Equations.

A reference for Eq.(3)
Eq.(3) in the previous manuscript was moved to Appendix Eq.(A1) in the revised one and a reference was given.

Errors in Eq.(4) and Eq. (5)
Eq.(4) and Eq. (5) in the previous manuscript was moved to Appendix Eq.(A1) and Eq. (A2) in the revised one and erroneous expressions in both the equations were corrected.

refer to Vionnet et al, 2012 instead of Vionnet et al, 2002
→corrected.

SSA( snow specific surface area)
We described SSA as one of shape or geometrical factors. (P11 L26) We also have an interest in SSA for one of important factors to describe snow microstructures.

References for an avalanche danger relating to the crystal type of snowfall
One reference was added. (P11 L28 and in references)

**Figures**
Fig.6 and Fig.7
A black contours were added around the mark of CMF points.

The correlation coefficient for the regressions in Fig.9 and others
The coefficient of determination $R^2$, which is square of the correlation coefficient is described

to each regression.

Fig 10: the differences between the two accumulation processes are not visually clear.
We eliminated Fig.10 in the previous manuscript as mentioned above.

**Language and spelling (text and figures)**
The submitted revised version (manuscript_Ishizaka_snowfall_density_R2.pdf) was checked and corrected by a native speaker.

**Text**
Plural form of nouns
We reexamined this problem and revised the manuscript.

Abstract L29 use "snowpack" instead of "snow pack"
→corrected.

Abstract L29 "practical use"
→corrected. (P1 L32)

P2 L21-22: the use of paragraph made of a single sentence….
We reexamined paragraphs in the previous manuscript and revised to avoid one-sentence paragraphs.

P3 L 8 use "snowfalls" instead of "snow falls"
→corrected.

P3 L 16 "through which snow falls and accumulates"
→corrected. (P4 L3)

P 3 L27 double use of "falling"
→corrected. (P4 L14)

P 5 L 8: the formulation "rather complicated situation" should be rephrased.
→revised. (P6 L7-9)

P 6 L 10-11: use "the size component" instead of "a size component"

→revised to different expression in the section 2.4. (P5 L16)

P7 L 21-22: the sentence "It is found …. for a event has" is complicated….
→revised (P7 L26-27)

P 9 L 6-7: use "snowpack" instead of "snow pack" (same for P9 L 21)
→corrected.

**Figures**

Caption of Fig. 9: add a space between kg and m$^{-3}$
→corrected.

References (not included in the previous manuscript)
We again appreciate your introduction of the references. They are very useful to improve our manuscript.

**Authors' responses to the Reviewer 2**

We would like to express our appreciation to the Reviewer #2 for your constructive reviews and helpful advice.
We have revised our manuscript and corrected mistakes according to your general and specific comments.

The revised manuscript file is "manuscript_Ishizaka_snowfall_density_R2.pdf".

Fist of all, the title has been slightly changed by following the editing process of a native speaker.
The new title is "Relationships between Snowfall Density and Solid Hydrometeors Based on Measured Size and Fall Speed, for Snowpack Modeling Applications"

Our responses to the comments are described as follows:

**Response to general comments 1**

Uncertainty of the approximated relationships
We recognize our observed data are not enough for statistical analysis. Our aim of this study is to establish a critical relationship between hydrometeor types and snowfall density, so we strictly selected the observation data at the event with almost the same type of hydrometeors. Although we carried out observations in three winter seasons and obtained 53 events, only 34 events could be used (other 19 event were not suitable for this study because they occurred under the complex precipitation condition). It was hard to increase sampling events. Therefore we could not obtain enough data to statistically analyze. For these reasons, to clear uncertainty of the relationships, we described the correlation efficient and standard errors for them in the revised manuscript. Furthermore, we estimated the density for snowfall events using the obtained relationships to examine applicability of our method in the section 3.4.1. The density estimated with the relationships approximately corresponds with observed density. This might be a sort of validation for the relationships.

**Response to general comment 2**

Comparison of our method to currently employed methods in numeric snowpack models

On the reviewer's instructions, we added the new section 3.4.1 in the revised manuscript to discuss applicability of our method. In the section, we presented comparison of estimated snowfall density using our method with that used the snowpack model developed by Lehning et al. (2002) and showed time series of observed and estimated densities.

**Responses on specific comments**

Uncertainty of scale
The scale used here have an uncertainty of about 0.2mm, with which error originates is fairly small and negligible. We described this information in the revised manuscript. (P5 L28; in the revised manuscript: same in below)

Paragraphs on densification error
On the reviewer's recommendation, we moved the paragraph that deviated the densification error to the Appendix in the revised manuscript.

Errors in Eq. (4) and Eq. (5)
Eq.(4) and Eq. (5) in the previous manuscript were moved to the Appendix Eq.(A1) and Eq. (A2) in the revised one and erroneous expressions in both the equations were corrected.

Section 3.1 "Classification of snowfall events" in the previous manuscript
On the reviewer's recommendation, the section was moved to the method section giving section no. 2.4 in the revised manuscript.

Applicability of our method to different snow types and long lasting events
Although we carried out short time observations to select almost the same type snowfall, we did not select typical aggregate case and graupel case arbitrarily. In our method, all type events are classified into the aggregate or graupel or two small particle groups, even though if it does not have typical characteristics of the classified type.
For long lasting event, we can estimate the density from each short interval density as mentioned in the section 3.4.1 combining measured mass flux. However for more accurate long term estimation, we should consider the effect of densification and metamorphosis to the accumulated snow at each interval. Moreover, we also should consider the effect of other meteorological effects to the accumulated snow. These effects should strongly depend on hydrometeors types. For these reasons, we think they are the remained issues for the future studies.

Figure Captions

On reviewer's instruction, we reexamined all Figure Captions and revised them. We again appreciate the reviewer's advice accompanying the specific example.

Specification of section in referring previous section

We described the section number in referring the previous section.

**Responses on technical corrections**

• P1 L31: plactical -> practical

→ corrected. (P1 L32)

• P2 L21-22: Try to eliminate one-sentence paragraphs.

We reexamined paragraphs in the previous manuscript and revised to avoid one-sentence paragraphs.

• P3 L12: "The winter temperature, around 0C..." Is this the mean winter temperature?

→revised as following:

"A temperature of around 0 °C during many snowfall events"   (P3 L28-29)

• P3 L28: "CCD" Please define this abbreviation.

→"CCD (Charge-Coupled Device)" (P4 L14)

• P8 L25: "SI" -> S1

→ corrected. (P9 L22)

• P10 Summary: The summary should be autonomous and abbreviations should be defined.

Abbreviations were defined in the summary.

• I recommend to rephrase the following sentences to make them more precise:

– P4 L19: "If different snow types, ..."

→revised (P5 L1-3)

-   P7 L21: "It is found that the densities..."

→revised (P7 L26-27)

• Missing/spare spaces:

→ All of them were corrected. Not listed individually.

Language

There are numerous lingual issues.

The submitted revised version (manuscript_Ishizaka_snowfall_density_R2.pdf) was checked and corrected by a native speaker.